# CAPYBARA: A generalizable framework for predicting serological measurements across human cohorts

Sierra Orsinelli-Rivers[1], Daniel Beaglehole[2], Tal Einav[1,3]*

**1** Center for Vaccine Innovation, La Jolla Institute for Immunology, La Jolla, California, United States of America, **2** Computer Science and Engineering, University of California San Diego, La Jolla, California, United States of America, **3** Department of Medicine, University of California San Diego, La Jolla, California, United States of America

\* tal.einav@lji.org

## Abstract

The rapid growth of biological datasets presents an opportunity to leverage past studies to inform and predict outcomes in new experiments. A central challenge is to distinguish which serological patterns are universally conserved and which are specific to individual datasets. In the context of human serology studies, where antibody-virus interactions assess the strength and breadth of the antibody response and inform vaccine strain selection, differences in cohort demographics or experimental design can markedly impact responses, yet few methods can translate these differences into the value±uncertainty of future measurements. Here, we introduce CAPYBARA, a data-driven framework that quantifies how serological relations map across datasets. As a case study, we applied CAPYBARA to 25 influenza datasets from 1997-2023 that measured vaccine or infection responses against multiple influenza variants using hemagglutination inhibition (HAI). To demonstrate how a subset of measurements in each study can infer the remaining data, we withheld all HAI measurements for each variant and accurately predicted them with a 2.0-fold mean absolute error—on par with experimental assay variability. Although studies with similar designs showed the best predictive power (*e.g.*, children data are better predicted by children than adult data), predictions across age groups, between vaccination and infection studies, and across studies conducted <10 years apart showed comparable 2–3-fold accuracy. By analyzing feature importance in this interpretable model, we identified global cross-reactivity trends that can be directly applied in future longitudinal or vaccine studies to infer broad serological responses from a small subset of measurements.

**Data availability statement:** Code is available through the accompanying GitHub repository (https://github.com/TalEinav/CAPYBARA). Analyses were implemented in Python using standard scientific libraries (NumPy, SciPy, scikit-learn).

**Funding:** This research was supported by LJI & Kyowa Kirin, Inc. (KKNA - Kyowa Kirin North America, TE), and the Bodman family (TE). The funders had no role in study design, data collection and analysis, decision to publish, or preparation of the manuscript.

**Competing interests:** The authors have declared that no competing interests exist.

## Author summary

The potential to integrate data from multiple studies is hampered by differences in cohort demographics or study design – such effects, even when known, are hard to estimate. Here, we analyzed 25 studies quantifying how influenza antibody responses inhibited different sets of viral variants. We developed a computational approach that learned which studies accurately predict one another and estimated the value and uncertainty of antibody inhibition against each variant. Although these studies were conducted over two decades, used different study designs, and assessed different age groups, each study was well-predicted by at least one other dataset, enabling rapid cross-dataset integration. This tool can be readily applied to future studies by unbiasedly quantifying study similarity based on prediction accuracy, or by measuring the antibody response against a small set of variants to predict the response against dozens of other variants, thereby leveraging the wealth of prior data to fuel future efforts.

## Introduction

As biological datasets continue to expand in size and complexity, it is becoming increasingly more challenging to integrate information from prior datasets to inform and predict the outcomes of future experiments. Patterns found in one group of individuals may not apply to another group where factors such as age, exposure history, or immune state differ [1–6]. More subtle, and often unknowable, differences in experimental methodology or batch effects may further affect which datasets can predict one another. While many studies have identified that cohorts differ in some way (*e.g.*, children and adults show significantly different immune responses [7–11]), we lack methods that estimate how these differences translate into future measurements. Such quantitative predictions are not only the hallmark of deeply understanding a system, but they also facilitate head-to-head comparisons across studies measuring different features.

This work tackles this problem in the context of the antibody response against the rapidly evolving influenza virus, which underpins the annual vaccine selection process [12,13]. Specifically, we consider serum hemagglutination inhibition (HAI) against multiple influenza variants, where higher HAI titers correlate with greater protection [14–16]. While thousands of new variants (or strains) emerge each year [17–19], only a small fraction can be functionally characterized using HAI, and the variants measured often differ between studies. Critically, we still lack methods that take a person's HAI titers against a few variants and infer their titers for other variants, which would quantify the holes in a population's immunity that should be closed when the vaccine is next updated.

Currently available HAI datasets have several direct clinical applications. Prior work has shown that a person's HAI against multiple strains can infer their influenza exposure history [20,21] or help predict their response to future vaccines [22].

Serum-virus HAI titers have been shown to be inherently low dimensional [23,24]. where titers against some variants can infer the titers of other strains [25,26]. As such, a new study seeking to measure HAI against numerous variants could theoretically extract these cross-reactivity relations from existing datasets, measure a minimal number of variants, and then predict the HAI of the remaining strains. One key hurdle is that cross-reactivity relations may differ with age, influenza exposure, and other immune variables. As the number of prior studies continues to increase, it is unclear *a priori* which datasets will best predict the cross-reactivity relations in another study, nor what form those relations will take.

To that end, we introduce the method Cross-study Adaptive Predictions Yielding Bayesian Aggregation with Recursive Analysis (CAPYBARA), a generalizable framework that efficiently selects the most predictive features within each dataset, determines their cross-reactivity relations, estimates prediction error, and then combines predictions from multiple studies weighted by their confidence. Fig 1 provides an overview of the CAPYBARA workflow, including the feature learning process, model training, error calibration, and Bayesian weighing of predictions across datasets. We demonstrate the utility of this approach by applying CAPYBARA to 25 influenza HAI datasets, providing a comprehensive analysis of cross-study prediction performance in a large-scale serology compilation. We first apply CAPYBARA to H3N2 data and then demonstrate its generalizability by predicting H1N1, B Victoria, and B Yamagata titers.

In the context of influenza immunity, CAPYBARA addresses two essential questions: First, how accurately can we leverage prior studies to predict future antibody inhibition data? Second, how few measurements are needed in order to extrapolate all antibody-virus interactions for any set of variants? Accurate cross-study predictions in the face of differences in study populations, experimental conditions, and virus panels would not only expedite future experiments but also help quantify the magnitude and breadth of the immune response in greater resolution.

## Results

### Overview of the algorithm

The CAPYBARA algorithm predicts the HAI titers of multiple sera against a withheld or unmeasured variant-of-interest $V_0$ in study-of-interest $S_0$. As input, we assume that HAI titers from other variants $V_1$, $V_2$, $V_3$… were measured for these same sera, and that other studies $S_1$, $S_2$… also measured HAI for $V_0$ and a subset of other variants (Fig 1A). Model features are the titers against different influenza variants, with HAI against $V_1$, $V_2$, $V_3$… used to predict the withheld HAI titer against $V_0$.

The algorithm proceeds as follows: 1) In every other study $S_j \in \{S_1, S_2… S_n\}$, identify the subset of variants (features) that best predict HAI titers for $V_0$ (Fig 1B). 2) Train a model in $S_j$ to predict each subject's titer ($\mu_j$) for $V_0$ (Fig 1B). 3) Repeat step 2 on all other variants $V_1$, $V_2$, $V_3$… whose values are known, so that within-study and cross-study error can be computed. This determines the error relationship when predicting from $S_j$ to $S_0$ (Fig 1C), which is then applied to determine the uncertainty $\sigma_j$ for $V_0$ predictions in $S_j$. 4) Combine predictions from all studies to estimate the HAI titer±error for each subject (Fig 1D, Methods). Prediction accuracy should only increase as more datasets are included, and adding a very noisy dataset ($\sigma_j \rightarrow \infty$) will negligibly change predictions. Multiple methods were tested for each model component (*e.g.*, random forest, ridge, lasso) on the Fonv studies, and the final CAPYBARA method is composed of Recursive Feature Machines, ridge regression, and Bayesian weighing that were the most accurate. Missing HAI entries were imputed using the row- and column-means of all measured values, yet all error metrics (*e.g.*, $\sigma_{Actual}$) were computed using the measured titers. Table 1 describes the four $\sigma$ terms used throughout this study.

To validate how well CAPYBARA predicted unmeasured serum-virus interactions across a compendium of influenza studies, we entirely withheld antibody responses from each variant within 20 vaccine studies and 5 longitudinal infection studies conducted between 1997–2023 (Table 2). These studies covered a variety of vaccine types (inactivated, live attenuated), age groups (children and adults), and geographic regions, containing ~200,000 HAI titers from 3,855 unique subjects (Table 2 and S1 Fig), Given this diversity, it was unclear *a priori* which datasets would be most informative to impute the HAI titers in any other study.

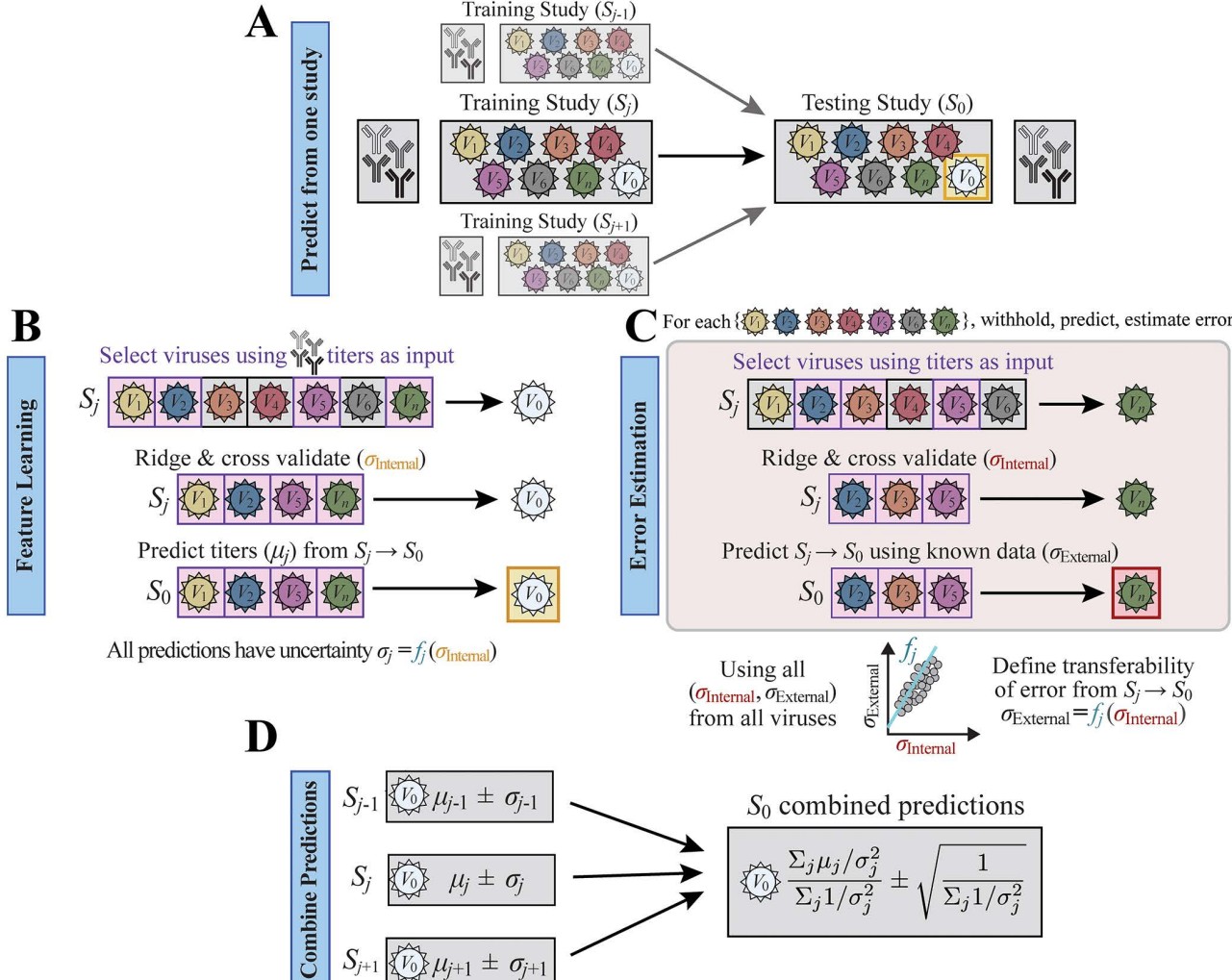

**Fig 1. Combining datasets using CAPYBARA to predict new antibody-virus interactions. (A)** Given studies (...$S_{j-1}$, $S_j$, $S_{j+1}$...) measuring serum HAI against a subset of influenza variants $V_0$-$V_n$, and study-of-interest $S_0$ measuring HAI against $V_1$-$V_n$, CAPYBARA predicts $V_0$'s measurements in $S_0$. **(B)** CAPYBARA first identifies the most predictive features (HAI against a subset of variants) using Recursive Feature Machines (pink boxes). Ridge regression is applied using those features, training on a subset of data in $S_j$ and cross-validating on the rest (error $\sigma_{Internal}$, Table 1). This model predicts titer values $\mu_j$ from $S_j \to S_0$ without uncertainty. **(C)** To estimate cross-study prediction error, every other variant is withheld and predicted from $S_j \to S_0$ to determine the internal ($\sigma_{Internal}$) and cross-study ($\sigma_{External}$) error. Combining the errors from every overlapping variant yields the transferability function $f_j$ that is applied to $V_0$'s $\sigma_{Internal}$ from Panel B to estimate the uncertainty $\sigma_j$ in $S_j$. **(D)** Predictions from all studies are combined through a Bayesian approach to yield a consensus prediction for the study-of-interest ($S_0$).

## Antibody responses are predicted between infection and vaccination studies within experimental noise

To test how well the HAI of new variants could be inferred across diverse biological contexts, we first examined how a longitudinal 6-year infection study (2007–2011 $Fonv_{Inf}$) predicted the titers of overlapping variants in a vaccine study conducted six years later (2017 $UGA_{Vac}$), by which point subsequent infections or vaccinations could have dramatically altered HAI cross-reactivity. In total, $N = 4{,}336$ titers were predicted in the vaccine study with root-mean-squared error (RMSE) $\sigma_{Actual} = 3.1x$ (where "x" denotes fold-error) between the predicted and measured titers (Fig 2A, green), implying that a measured HAI $= 20$ will typically be predicted as a titer between $20/3.1 = 6.5$ and $20 \cdot 3.1 = 62$. The model's estimated error

**Table 1. Definitions of CAPYBARA error terms and their roles in model training, transferability, and evaluation. Models are trained to infer HAI titers for variant $V_0$ in study $S_j$ and then applied to predict $V_0$'s titers in study $S_0$. Titers from other variants $V_k$ can be chosen as model features.**

| Symbol* | Description | How it is computed | Where it is used |
|---|---|---|---|
| $\sigma_{Internal}$ | *Within-study* prediction error in $S_j$ | RMSE on held-out subjects during internal cross-validation (train 80%, test 20%) | *x*-axis in transferability function to compute $\sigma_{Predicted}$ from $S_j \to S_0$ |
| $\sigma_{External}$ | *Cross-study* prediction error in $S_0$ | By predicting every other virus $V_k$ in $S_0$ (while withholding $V_0$) | *y*-axis in transferability function to compute $\sigma_{Predicted}$ from $S_j \to S_0$ |
| $\sigma_{Predicted}$ | Estimated prediction error in $S_0$ | Using $\sigma_{Internal}$ and $\sigma_{External}$ from all other $V_k$'s, and $\sigma_{Internal}$ from $V_0$ | Compute for each $S_j$, then combine using Bayesian weighing (weight $\propto 1/error^2$) |
| $\sigma_{Actual}$ | Ground truth prediction error in $S_0$ | RMSE between predicted and measured titers for $V_0$ | Determine prediction accuracy |

*All RMSEs are computed on $\log_2(HAI/5)$ and then unlogged through exponentiation by 2.

**Table 2. List of large-scale influenza studies used in this analysis. 25 influenza datasets comprising vaccine [Vac, white background] or infection studies [Inf, gray background] used to assess cross-study predictions. The year represents when each study was conducted (*e.g.*, 2010-2014 implies that samples were collected annually across these 5 years). Sera collected at different time points from the same subject were considered independently. The total number of measurements in each study equals (# of sera)×(# of viruses)-(% missing).**

| Year + Name of Study | # of Measurements | # of Sera | # of Viruses | % Missing | Vaccine Type |
|---|---|---|---|---|---|
| 1997 Fonv$_{Vac}$ [27] | 13,611 | 212 | 69 | 7.0% | Inactivated (Afluria trivalent) |
| 1998 Fonv$_{Vac}$ [27] | 16,180 | 256 | 69 | 8.4% | Inactivated (Afluria trivalent) |
| 2009 Fonv$_{Vac}$ [27] | 3,186 | 160 | 20 | 0.4% | Inactivated (Afluria trivalent) |
| 2010 Fonv$_{Vac}$ [27] | 3,165 | 160 | 20 | 1.1% | Inactivated (Afluria trivalent) |
| 2010 Ert$_{IIV,Vac}$ [28] | 1,148 | 82 | 14 | 0% | Inactivated (Influvac or Vaxigrip trivalent) |
| 2013 Ert$_{IIV,Vac}$ [28] | 1,078 | 77 | 14 | 0% | Inactivated (Influvac or Vaxigrip trivalent) |
| 2012 Ert$_{LAIV,Vac}$ [28] | 1,288 | 92 | 14 | 0% | Live attenuated (Fluenz trivalent) |
| 2013 Ert$_{LAIV,Vac}$ [28] | 1,512 | 108 | 14 | 0% | Live attenuated (Fluenz trivalent) |
| 2014 Hin$_{V,Vac}$ [29] (Vaccinated or infected) | 1,081 | 69 | 16 | 2.1% | Inactivated (Fluzone trivalent) or Live attenuated (FluMist quadrivalent) |
| 2015 Hin$_{V,Vac}$ [29] (Vaccinated or infected) | 1,286 | 82 | 16 | 2.0% | Inactivated (Fluzone trivalent) |
| 2015 Hin$_{U,Vac}$ [29] (Unvaccinated and uninfected) | 992 | 62 | 16 | 0% | Inactivated (Fluzone trivalent) |
| 2016 Fox$_{Nam,Vac}$ [5] (Ha Nam) | 20,688 | 597 | 36 | 3.7% | Inactivated (Vaxigrip trivalent) |
| 2016 Fox$_{HCW,Vac}$ [5] (Health Care Workers) | 4,689 | 147 | 32 | 0.3% | Inactivated (Fluarix quadrivalent) |
| 2016 UGA$_{Vac}$ [30] | 5,032 | 296 | 17 | 0% | Inactivated (Fluzone trivalent) |
| 2017 UGA$_{Vac}$ [30] | 9,756 | 542 | 18 | 0% | Inactivated (Fluzone trivalent) |
| 2018 UGA$_{Vac}$ [30] | 5,000 | 500 | 10 | 0% | Inactivated (Fluzone trivalent) |
| 2019 UGA$_{Vac}$ [30] | 7,376 | 922 | 8 | 0% | Inactivated (Fluzone trivalent) |
| 2020 UGA$_{Vac}$ [30] | 4,746 | 678 | 7 | 0% | Inactivated (Fluzone quadrivalent) |
| 2021 UGA$_{Vac}$ [30] | 4,718 | 674 | 7 | 0% | Inactivated (Fluzone quadrivalent) |
| 2023 Crotty$_{Vac}$ [22] | 336 | 48 | 7 | 0% | Inactivated (Afluria quadrivalent) |
| 2007-2011 Fonv$_{Inf}$ [27] | 4,748 | 133 | 37 | 3.5% | Infection Study |
| 2007-2012 Fonv$_{Inf}$ [27] | 5,856 | 161 | 37 | 1.7% | Infection Study |
| 2009-2011 Hay$_{Inf}$ [31] | 20,905 | 1046 | 20 | 0% | Infection Study |
| 2012-2015 Hay$_{Inf}$ [31] | 24,240 | 1212 | 20 | 0% | Infection Study |
| 2010-2014 Ert$_{Inf}$ [28] | 504 | 36 | 14 | 0% | Infection Study |

$\sigma_{Predict}$ = 7.7x represents an upper bound (worst case) error, and although this bound was not tight, it satisfied $\sigma_{Actual} \lesssim \sigma_{Predict}$ as expected. In contrast, when we predicted this same 2017 $UGA_{Vac}$ dataset using a vaccine study from one year earlier (2016 $UGA_{Vac}$), we found a smaller prediction error $\sigma_{Actual}$ = 2.0x and a tighter estimated error $\sigma_{Predict}$ = 2.3x when predicting these same $N$ = 4,336 titers (Fig 2A, blue).

For each serum-virus HAI, the estimated titer and error ($\mu_1 \pm \sigma_1$ from study 1, $\mu_2 \pm \sigma_2$ from study 2) were combined using Bayesian statistics, $(\mu_1/\sigma_1^2 + \mu_2/\sigma_2^2)/(1/\sigma_1^2 + 1/\sigma_1^2) \pm (1/\sigma_1^2 + 1/\sigma_2^2)^{-1/2}$, which places more weight on the more confident prediction with smaller $\sigma_{Predict}$ (**Methods**). In this case, 2016 $UGA_{Vac}$ was weighed ~20x more heavily ($1/\sigma_1^2$ = 0.19 vs $1/\sigma_2^2$ = 0.01), as may be expected from its similar study design. The combined predictions remained as good as the predictions from the 2016 $UGA_{Vac}$ study alone ($\sigma_{Actual}$ = 2.0x, $\sigma_{Predict}$ = 2.1x), demonstrating that the model is not hampered by adding the poorly predicted infection study (Fig 2A, purple).

As another example, we used an infection study (2010–2014 $Ert_{Inf}$) and a vaccine study (1997 $Fonv_{Vac}$) to both individually and jointly predict another infection study (2007–2011 $Fonv_{Inf}$). Interestingly, predictions between the infection→infection study ($\sigma_{Actual}$ = 3.1x) were very slightly worse than vaccine→infection predictions ($\sigma_{Actual}$ = 2.9x), even though the infection studies overlapped in time while the vaccine study occurred ten years earlier (Fig 2B, green/blue). Combining both studies led to more accurate predictions than either dataset alone ($\sigma_{Actual}$ = 2.4x) with similarly tight estimated error ($\sigma_{Predict}$ = 2.5x, Fig 2B, purple).

More generally, the predictions from any number of datasets can be combined using error estimation (Methods). As representative examples, we used every dataset in Table 2 to predict HAI titers in an adult health care workers vaccine study (2016 $Fox_{HCW,Vac}$, Fig 2C), vaccinated children (2014 $Hin_{V,Vac}$, Fig 2D), and an adult infection study (2012–2015 $Hay_{Inf}$, Fig 2E; all studies in S2 Fig). The $10^3$-$10^4$ predicted HAIs in each study had low RMSE ($\sigma_{Actual}$ = 1.8-2.1x, similar to the error of the HAI assay) and comparably low estimated error ($\sigma_{Predict}$ = 1.4-1.7x), demonstrating that combining datasets can precisely and confidently extrapolate HAI titers for completely unmeasured variants. These results corroborate that studies do not need to be pre-screened, since the framework will identify the most predictive datasets and ignore the poorly predictive ones.

**More training datasets lead to better prediction accuracy across 25 years of data**

To test generalizability, we compared the prediction error when training on any single dataset vs the combined predictions from all studies (Fig 3A). As expected, the individual datasets showed far larger variability ($\sigma_{Actual}$ = 1.6-10.7x) than the combined predictions (1.7-2.5x) that were always comparable to the most accurate pairwise predictions in each column. Interestingly, prediction accuracy does not need to be symmetric. For example, multiple studies had poor predictions with $\sigma_{Actual}$ > 6x when trained solely on 1997 $Fonv_{Vac}$, yet using any training dataset to predict values in 1997 $Fonv_{Vac}$ led to more accurate $\sigma_{Actual}$ < 4x. Computing signed prediction error showed that predictions were skewed to be slightly larger when the measured titer = 5 and slightly smaller when the measured titer was for higher ≤640, yet the median predictions were within 2-fold for measured titers ≤1280 and within 4-fold for larger titers (S3 and S4 Figs).

The greatest signed error deviations occurred when the two 2012–2013 $Ert_{LAIV,Vac}$ studies were used for training where, unlike in all adult studies, HAI titers for nearly all viruses hit the limit of detection (HAI = 5), leading to markedly different cross-reactivity relations. Even so, the combined predictions using all studies uniformly had signed error ≈0, demonstrating improved accuracy when using more datasets (S4 Fig).

At the individual-person level, there was a noticeably greater spread in pairwise predictions ($\sigma_{Actual}$ = 2.6x across all subjects, Fig 3B) than in the combined predictions (2.0x, Fig 3C), with 14.3% of the former predictions having an error > 4x while only 5.3% of the latter predictions had such error (S5 Fig). Indeed, CAPYBARA does better than averaging the individual predictions from each study by heavily weighing the more reliable, and hence more accurate, predictions (S6 Fig).

A few general trends can be seen from pairs of studies that poorly predict one another (Fig 3A). The two oldest studies (1997/1998 $Fonv_{Vac}$) tend to poorly predict studies from 2010 and beyond. The LAIV studies (2012/2013

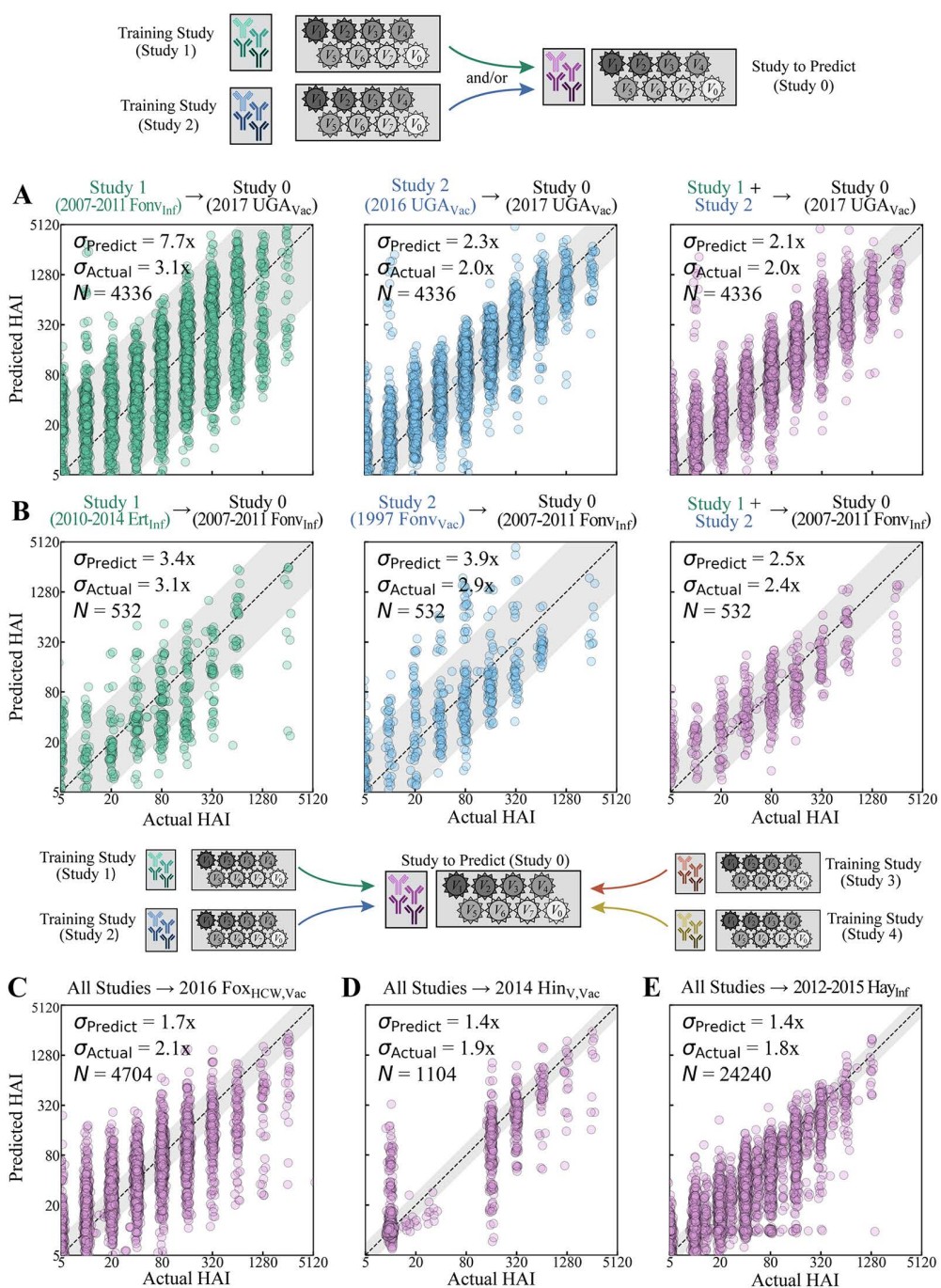

**Fig 2. HAI titers across vaccination and infection studies are consistently predicted within experimental noise by combining predictions from all other studies.** (A,B) Example predictions trained on an individual dataset (left and middle columns) and the combination of both datasets (right column). Labels above each plot identify the training→testing datasets. (C-E) Predicting three datasets using all other studies in Table 2. The estimated fold-error ($\sigma_{Predict}$), measured fold-error ($\sigma_{Actual}$), and the number ($N$) of predicted titers are shown, with the gray diagonal bands representing $\sigma_{Predict}$.

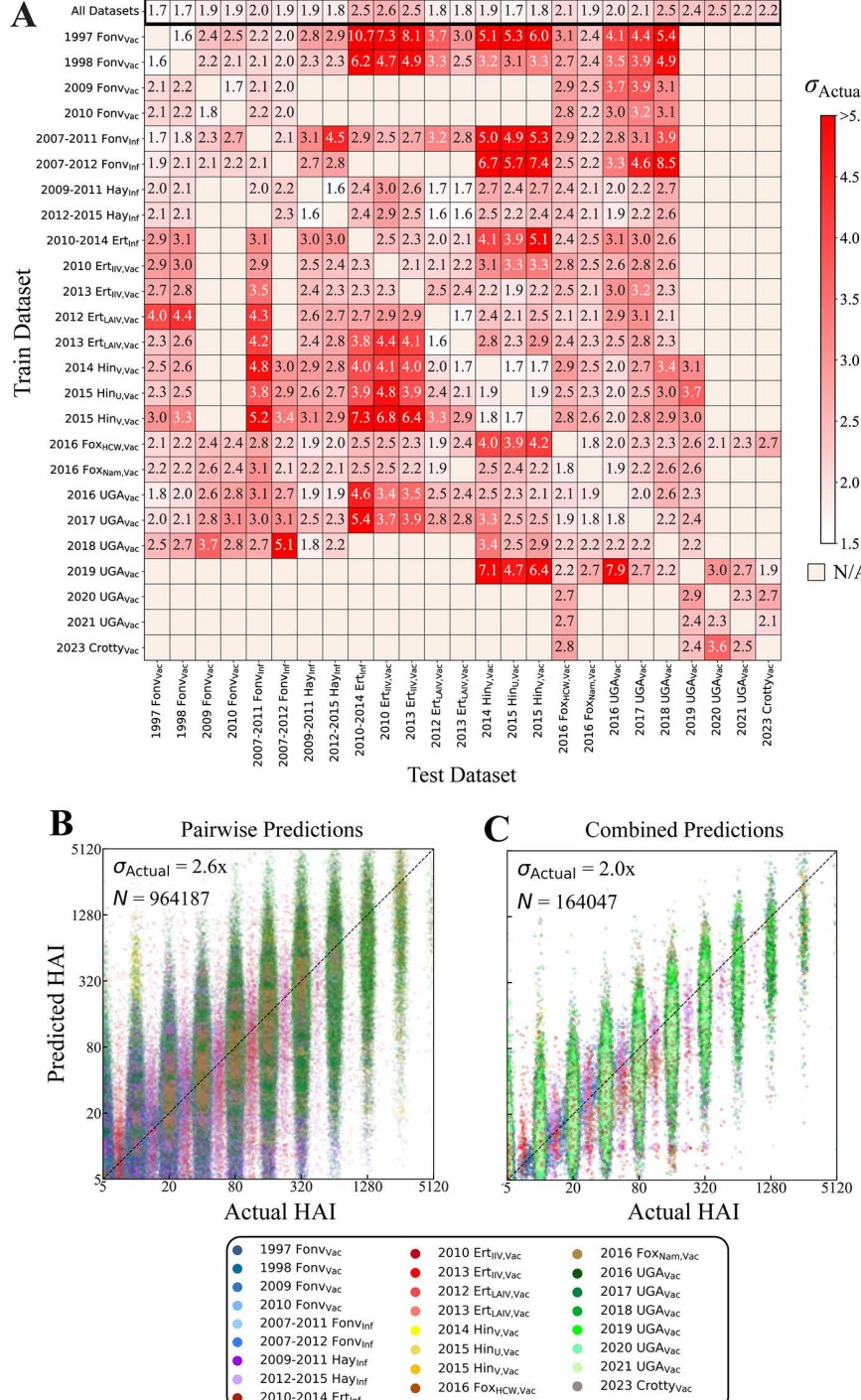

**Fig 3. Predicting HAI responses across all studies. (A)** Heatmap of the average RMSE ($\sigma_{Actual}$) across all subjects and overlapping variants in a study-of-interest (column). Training is either done using all studies (top row) or using a single study (all other rows). **(B-C)** All predicted versus measured HAIs when training on (B) a single study or (C) all other studies. The number $N$ of predictions is larger for pairwise predictions since the same serum-virus pair is predicted multiple times using different training datasets. The diagonal line $y = x$ represents perfect predictions.

Ert$_{LAIV,Vac}$) were sometimes poorly predicted by the more common IIV studies. Beyond these few rules, it was often unclear which studies would poorly predict one another, emphasizing the utility of CAPYBARA to infer such relationships directly from the data.

To demonstrate the generalizability of this approach, we next used CAPYBARA to predict all H1N1, B Victoria, and B Yamagata HAI titers in these same studies. Although the virus panels were smaller in each case, the UGA studies had the necessary overlap of ≥3 variants. Combined predictions of HAI titers using all datasets led to $\sigma_{Actual}$ = 1.9-2.6x for H1N1, comparable to H3N2 prediction accuracy (S7 Fig). B Victoria and B Yamagata achieved $\sigma_{Actual}$ = 1.7-3.7x, with most studies against being predicted with ≈2-fold error (S7 Fig).

Lastly, we showed that this CAPYBARA can disregard an extremely noisy dataset by generating a new study with random H3N2 HAI titers drawn from the same distribution and with the same variants as the 2016 UGA$_{Vac}$ and 2007–2011 Fonv$_{Inf}$ studies (S8 Fig). Adding this noisy dataset to the existing studies did not affect overall prediction accuracy, as expected. Thus, more datasets can be added until the desired prediction accuracy is achieved. Systematic shifts (e.g., all titers increased by 4x) are automatically accounted for, since HAI titers for one variant are inferred using titers from other variants within that same study (S9 Fig).

### Subsetting datasets helps explain prediction dynamics

Since age is well known to affect the antibody response, we assessed how well children (age ≤ 18) can predict adult responses (age > 18) and vice versa. Datasets were categorized as containing children only, adults only, or a combination of both (S1 Fig). HAI titers from studies in each category were exclusively predicted using models from either the same or a different category (Fig 4A). As expected, the best predictions came from models trained within the same category. For example, children's titers were better predicted by children data ($\sigma_{Actual}$ = 1.7x) than by adult data ($\sigma_{Actual}$ = 2.4x; $p < 0.05$, two-sided permutation test). Studies containing both children and adults represented an intermediate phenotype, which was itself best predicted by studies containing titers from children and adults. Despite most of these age effects being significant, the absolute effect of age was small, where even purposefully mismatching datasets (predicting children→adults or adults→children) led to median error < 2.5x.

We next split datasets by study type (vaccine versus infection). As before, there was a small but significant improvement in prediction accuracy when the same type of dataset was used for training (Fig 4B). For example, predicting from infection→infection studies ($\sigma_{Actual}$ = 1.7x) was more accurate than vaccination→infection ($\sigma_{Actual}$ = 2.0x; $p < 0.05$, two-sided permutation test), although predictions in either case were surprisingly accurate, with similar results when predicting vaccination responses.

The worst prediction accuracy was seen when splitting datasets by their year of study and using old datasets to predict responses >10 years into the future (Fig 4C). Studies were binned in five year increments, with studies conducted over multiple years represented by their median year. Training on studies from the same bin either led to the best predictions or to comparable predictions with the best bin (median $\sigma_{Actual}$ 1.6–2.1x). Measured accuracy dropped, often significantly, when using older datasets to predict more recent ones. In particular, training the oldest 1996–2000 datasets led to poor predictions and large variation on 2011–2015 ($\sigma_{Actual}$ = 3.2x; $p < 0.05$, two-sided permutation test) or 2016–2020 data ($\sigma_{Actual}$ = 2.6x; $p < 0.05$, two-sided permutation test), although predictions going backwards in time by >10 years tended to be more accurate. Estimated accuracy ($1/\sigma_{Predicted}^2$) showed similar behavior, with larger confidence for studies conducted within 10 years of one another (S4 Fig).

Lastly, we examined how accurately pre-vaccination titers predicted the peak post-vaccination titers (21–43 days post-vaccination) across vaccine studies. Surprisingly, we observed nearly identical prediction accuracies (median within ~0.02x of each other; p = 1.0, two-sided permutation test), suggesting that the HAI cross-reactivity across variants holds over time, with most variants increasing in tandem post-vaccination.

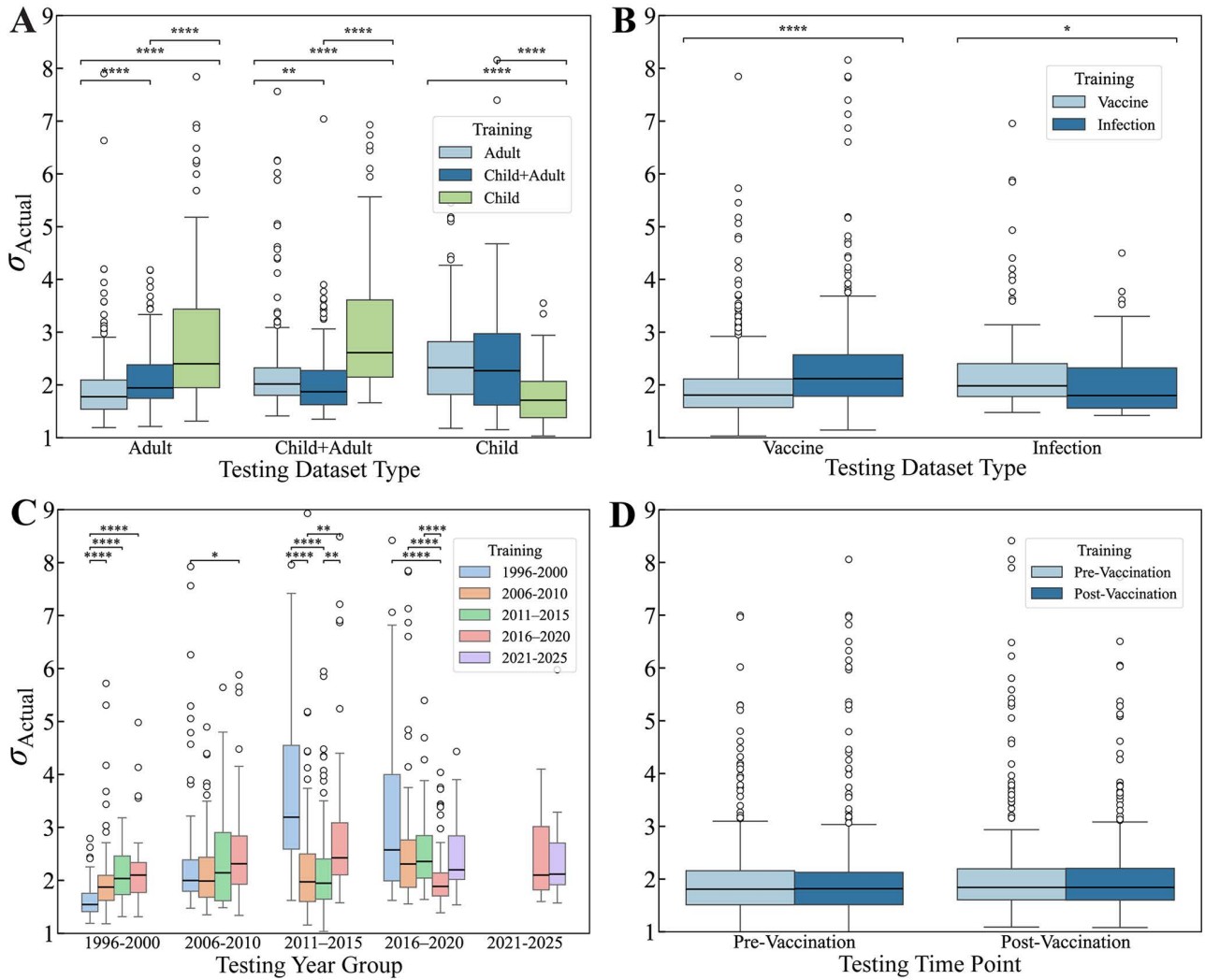

**Fig 4. Training on similar datasets marginally improves prediction accuracy.** Cross-study RMSE ($\sigma_{Actual}$) when training and predicting between datasets based on (A) the age groups adult-only, children-only, or mixed (child+adult); (B) vaccination or infection studies; (C) datasets grouped in 5-year intervals based on their median year; or (D) pre-vaccination (Day 0) vs post-vaccination (~1 month) data. Each box plot shows the distribution of errors for all possible withheld variants. The horizontal line denotes the median, boxes show the interquartile range, and whiskers extend to 1.5 times the interquartile range. Circles denote outliers. Statistical significance was assessed using two-sided permutation tests with Benjamini–Hochberg correction for multiple testing. Asterisks denote adjusted $p$-values: $**** = p < 0.0001$, $*** = p < 0.001$, $** = p < 0.01$, $* = p < 0.05$.

## Identifying universal relations between influenza variants

To demonstrate how future studies can leverage CAPYBARA to measure a few variants and infer the response from others, we sought universal relations that could be applied to a new study without requiring dataset reweighing through CAPYBARA. To that end, we used RFM to denote which variant features were the most important when predicting each of the 112 variants across these studies (Figs 5A and S10; red represents greater importance).

While variants circulating more than 10 years apart could be important (average feature importance = 0.3), the most important variant pairs tended to circulate less than a decade apart (average feature importance = 0.5, S10 Fig). However, feature importance could only be determined when two viruses were measured in at least two studies, so that the more

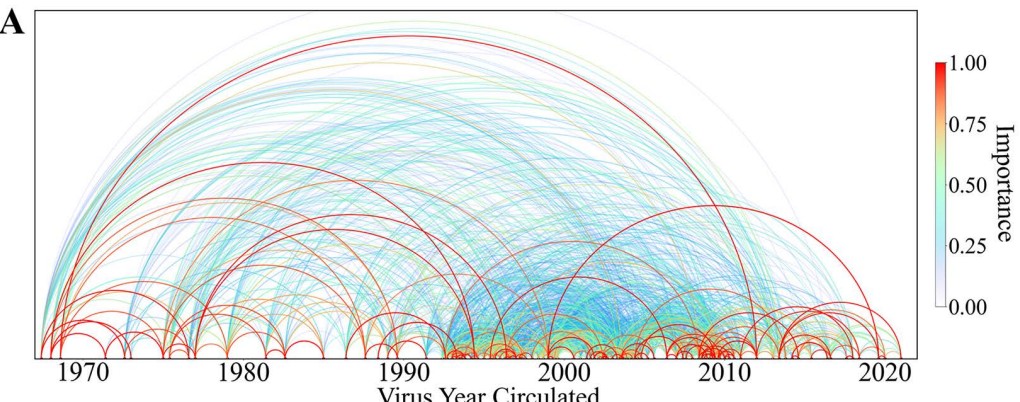

**B**

| Target Virus | 1 Feature Equation | 2 Feature Equation |
|---|---|---|
| Darwin 2021 | -0.63 + 0.44·Hong Kong 2019 | -1.65 + 0.28·Hong Kong 2014 + 0.55·Singapore 2016 |
| Tasmania 2020 | | 1.21 + 0.47·Hong Kong 2019 + 0.40·Hong Kong 2014 |
| Hong Kong 2019 | 0.68 + 0.80·Tasmania 2020 | -0.05 + 0.28·Panama 1999 + 0.60·Singapore 2016 |
| Kansas 2017 | | 0.25 + 0.44·Darwin 2021 + 0.48·Singapore 2016 |
| Singapore 2016 | 0.09 + 0.92·Hong Kong 2014 | -0.42 + 0.45·Hong Kong 2014 + 0.63·Perth 2009 |
| Hong Kong 2014 | 0.69 + 1.01·Perth 2009 | 0.11 + 0.53·Perth 2009 + 0.26·Texas 2012 |
| Switzerland 2013 | | -3.55 + 0.29·Perth 2009 + 1.05·Texas 2012 |

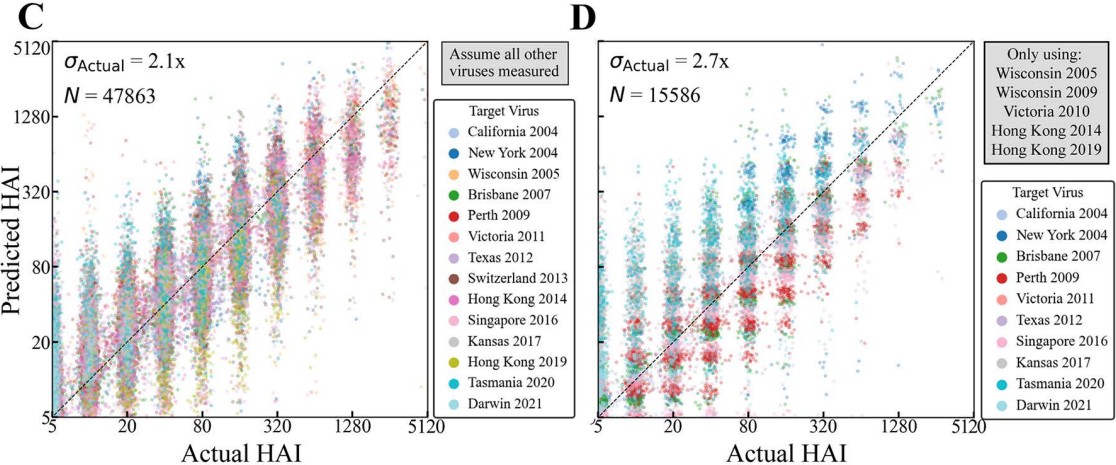

**Fig 5. A global dictionary of influenza variant importance. (A)** Rainbow diagram of feature importance between any pair of variants (connections are bidirectional). **(B)** Examples of universal HAI titer equations for multiple influenza vaccine strains, using titers from one variant (when possible) or two variants. Each virus name stands for its $\log_2(HAI/5)$ titer. See S1 File for all relations using ≤5 variants. **(C)** Measured versus predicted HAI titers for all vaccine strains in each study. Predictions were averaged from all other studies that measured the necessary variants. **(D)** Example using a small subset of five variants to predict ten other vaccine strains.

frequently selected vaccine strains tend to have far better coverage than non-vaccinate strains. For example, the 1968 pandemic strain Hong Kong 1968 was often measured, and it exhibited strong feature importance of ≈1 against viruses circulating as late as Hong Kong 2014.

Each variant-of-interest $V_0$ in study $S_0$ is predicted by every other study with at least three overlapping strains, leading to multiple potential distinct HAI relations. While 53.1% of relations required 1–2 variants, 20.3% of equations required 4

or more variants (all relations shown in S1 File). Since vaccine strains were frequently measured, many relations exclusively use these strains (examples in Fig 5B).

To evaluate the accuracy of these relations, each vaccine strain's HAI titers were individually withheld from a study-of-interest and derived by averaging the relations from all other studies. To make these results as generalizable as possible, predictions were not weighted by their estimated accuracy as in the sections above, but instead averaged equally across all studies. The resulting predictions showed an RMSE of 2.1x, comparable to the ≈ 2-fold error of the HAI assay, provided that each study measured all of the necessary variants to apply these relationships (Fig 5C).

To further expedite future studies, we assessed whether measuring a smaller set of only five influenza variants (comprising four vaccine strains and one non-vaccine strain) could predict ten other vaccine strains (Fig 5D and S1 File). This reduced set of variants only had a slightly larger RMSE of 2.7x, showing that cross-study relationships can increase the amount of data generated by a few experiments, and that prediction accuracy should increase as more datasets are measured, or by applying CAPYBARA to heavily weigh the most accurate studies.

Finally, we determined the key HA sites that underlie influenza cross-reactivity in two different ways. Method #1 identified the smallest number of HA1 amino acids whose edit distance best matches the RFM-derived virus-virus similarity (a.k.a. *Minimal Key Residues*). Starting from the full HA1, we eliminated one amino acid at a time that led to the greatest similarity (smallest Frobenius norm) between virus edit distance at the remaining sites and RFM importance. This identified a small set of 31 HA1 positions, where the virus-virus edit distance at these sites (S11A Fig) roughly reproduced their RFM-derived feature importance (Fig 5A) with Pearson correlation $r = 0.60$. In both methods, we found the expected similarity between contemporaneous variants as well as between some variants circulating 10+ years apart.

Method #2 identified the key conserved residues across distant variants necessary for accurate predictions (a.k.a. *Conserved Sites in Distant Variants*). Thus, we analyzed the 11 pairs of variants circulating 10+ years apart with ≥0.7 RFM importance. Taking the most common conserved sites in >80% of pairs identified 167 amino acids, with this larger number arising because the 11 pairs explored a limited part of sequence space. Across all variants, the edit distance at these sites was at most 6, demonstrating that these are highly conserved sites in HA1 (S11B Fig).

When we compared the residues found from each method, we found that 21 of them overlapped, representing the positions most likely to be important for both variants circulating in similar years as well as 10+ years apart (S11C and S11D Fig). As a further check, 14 residues fell within the canonical H3N2 epitopes A-E and the receptor binding site, highlighting their biological plausibility. 4 other sites lie outside these canonical sites but have been previously reported to impact H3N2 antigenicity, while 3 additional sites outside the canonical epitopes were identified by this analysis [5,32–35]. Taken together, the HAI data suggest that these 21 sites are responsible for both the short-range and long-range similarity between influenza variants.

## Discussion

Here, we developed CAPYBARA, a general algorithm that combines feature learning, model generation, and error estimation to predict unmeasured interactions based on existing datasets. As a case study, CAPYBARA was applied to identify universal patterns in serum-virus cross-reactivity and predict each serum's HAI against variants that were entirely withheld from a study. While factors such as age [7–10,36] or exposure history [1,3,5,6] are known to affect the antibody response, it is unclear how these impact serum cross-reactivity. To that end, CAPYBARA quantifies how accurately the local relationships in one dataset translate into another dataset using all non-withheld data, testing this approach across 25 different influenza studies. A key innovation from previous methods is that this model combines state-of-the-art feature selection [37] and error estimation techniques [38] while leveraging ridge regression for greater interpretability. By using predictions of overlapping variants to estimate prediction error between studies, this approach unbiasedly determined which studies exhibit the same cross-reactivity relations directly from the data. Interestingly, there was always at least one study with accurate predictions, and hence combined predictions trained on all datasets were uniformly accurate with 1.7-2.5x

prediction error. The resulting cross-reactivity relations could be partially recouped by a subset of 21 amino acids in the HA head, offering a direct link between model-derived relationships and underlying sequence features.

Subject age had a small but significant effect, suggesting that cross-reactivity changes from childhood (age ≤ 18) into adulthood (age > 18). Children predicted other children's responses better than adults, while adults predicted other adult responses better than children, with mixed datasets containing both children and adults falling in the middle. The year a study was conducted also had a significant effect, with studies within a 10 year window exhibiting 1.6x-2.4x error while studies done further apart in time had 2.0x-3.2x error. However, studies conducted within 10 years were not always highly predictive, and future studies should explore what other study features lead to similar cross-reactivity relations and better predictive power. Vaccination and infection studies similarly predicted their own category better than the other category. Surprisingly, within vaccine studies, the pre-vaccination (day 0) and peak response (day 21–40) time points predicted one another with comparable accuracy, suggesting that pre- and post-vaccination cross-reactivity resemble one another. This could arise if all variant HAIs increase by a similar amount post-vaccination, or if post-vaccination responses are relatively weak, both of which held true across these datasets and were previously reported [39].

One limitation of this approach is that a variant's HAI titers can only be predicted in a dataset-of-interest if that variant has been measured in at least one other study. Thus, this method is not equipped to predict the HAI of new variants, although a variant measured in one dataset can be predicted in all other studies. As datasets measuring more variants are added, the number of predictions in each study grows combinatorially.

As such, CAPYBARA lays the foundation to design more efficient experiments that leverage existing studies. It further provides a quantitative foundation to determine the minimum number of variants that should be measured to infer the HAIs from multiple variants of interest. To facilitate such use, we also provide the average cross-reactivity relations between all H3N2 influenza variants examined in this work (S1 File). These relations can be immediately applied to a new study, or they can be further augmented with CAPYBARA that will derive new dataset-specific relations weighed by dataset similarity.

In principle, CAPYBARA could help augment vaccine strain selection by unifying the insights gained from influenza surveillance from around the world. For example, each year the WHO Collaborating Centers measure how human sera from their country inhibit potential vaccine strains from their region, resulting in partially overlapping virus panels that are local to each site. If each site agrees to measure the same three viruses (*e.g.*, the most recent three vaccine strains), they could immediately infer how sera measured at any WHO site would inhibit each variant measured at every other site, thereby quantifying the potential for viral escape across the world.

## Methods

### Overview of the datasets

We analyzed a collection of 25 influenza vaccine and infection studies spanning 1997–2023 (Table 2). If one participant had multiple sera (*e.g.*, pre-vaccination and post-vaccination), the two were analyzed independently. Predictions were carried out between two datasets if they measured HAI against at least three of the same H3N2 variants, since this ensures that there are enough features for cross-study prediction.

To assess generalizability, CAPYBARA was applied without modification to HAI titers for H1N1, B Victoria, and B Yamagata in these same datasets. Each subtype or lineage was predicted independently, with model fitting, cross-dataset prediction, and error estimation performed identically to the H3N2 analysis.

### Analyzing HAI titers

All studies used hemagglutination inhibition, which measures the highest dilution of serum at which hemagglutination is inhibited. A larger HAI titer will reflect a more potent serum, but it may also reflect differences in virus passaging (egg- vs

cell-grown) or study design (incubation conditions, type or batch of red blood cells). Missing HAI titers, comprising 2.1% of all measurements, were imputed using the row–column mean since both RFM and ridge regression require complete data. While these imputed values were used for model training, model evaluation was only carried out on the measured values.

As in prior analyses, titers were transformed to $\log_2(\text{HAI}/5)$, which reduces the bias toward large titers [14,38]. All prediction errors are shown in unlogged units so that they can be compared to the measured HAI titers. More precisely, the root-mean-squared error ($\sigma_{\text{Actual}}$) of the logged titers is exponentiated by 2 to get the unlogged error (*i.e.*, $\sigma_{\text{Actual}} = 1.0$ for $\log_2$ titers corresponds to an error of $\sigma_{\text{Predict}} = 2^{1.0} = 2$-fold, with "fold" or "x" indicating an un-logged number). Prior work has shown that the HAI has an inherent 2-fold error on average [22], and hence predictions with ≈2-fold error are as accurate as possible. Batch correction across studies was not applied, since HAI titers are standardized using reference antisera to maintain comparability across experiments.

We tested if including demographic covariates (age, sex, BMI) as additional features in the ridge regression improved prediction accuracy. Sex and BMI led to comparable prediction accuracy. Age led to far *worse* prediction accuracy when extrapolating from adults→children or children→adults studies, since linear regression is poorly suited to such large extrapolations. However, when combining predictions from all studies, even with age information, CAPYBARA up-weighed the more accurate children→children or adult→adult predictions and retained ≈2-fold prediction error in every study. Although demographic information was removed during the analyses above, these results demonstrate that titers can be robustly predicted even when adversarial information is added, provided that at least some studies yield accurate predictions.

A synthetic noisy dataset was created by randomly sampling HAI titers from the joint distribution of real training studies (2016 UGA$_{\text{Vac}}$ and 2007–2011 Fonv$_{\text{Inf}}$), preserving the same virus panel and assuming the average number of subjects between the two studies (214 subjects).

## Overview of CAPYBARA

We first outline the four main steps of the algorithm and then describe each in detail:

**Step 1: Feature Learning** (Fig 1B): For each external study $S_j$ that measured the target virus $V_0$, a Recursive Feature Machine [37] identifies a small subset of variants that best predict $V_0$.

**Step 2: Model Training** (Fig 1B): Ridge regression is applied to a subset of sera within $S_j$, using the selected variants as inputs and $V_0$ as the output. The internal root-mean-square error $\sigma_{\text{Internal}}(V_0)$ is computed on the withheld sera.

**Step 3: Cross-Study Error Calibration** (Fig 1C): To extrapolate the regression relation from $S_j$ (where $\sigma_{\text{Internal}}$ is known) to the new dataset $S_0$, CAPYBARA withholds each variant $V_k \neq V_0$ and measures how its internal error in $S_j$ maps to its external error $\sigma_{\text{External}}(V_k)$ in $S_0$. A piecewise linear function is then fit to the ($\sigma_{\text{Internal}}$, $\sigma_{\text{External}}$) pairs for all $V_k$, and this function is applied to $\sigma_{\text{Internal}}(V_0)$ to estimate the error in $S_0$, denoted by $\sigma_{\text{Predict}}(V_0)$.

**Step 4: Combined Predictions** (Fig 1D): When multiple studies can predict a virus $V_0$ in $S_0$, their predictions are combined using Bayesian weighting, *i.e.*, weighting each prediction inversely by its squared predicted error, $(1/\sigma_{\text{Predict}})^2$. This yields a single predicted HAI titer and a calibrated uncertainty estimate for that titer.

CAPYBARA was designed to balance simplicity, efficiency, and interpretability. RFM and ridge regression ensure that the minimum number of predictive features are used, although each component could be modified independently (e.g., using random forests instead of ridge). The resulting CAPYBARA architecture was chosen after testing a number of different architectures, with the current method optimizing speed and accuracy.

Predictions between two studies are carried out independently of any other studies. Hence, studies can be weighed and combined in a modular fashion, and introducing a new study does not require rerunning the model on prior datasets. A new study's predictions for a study-of-interest can be directly combined with all prior predictions using Bayesian inverse-variance weighing (as in Step 4).

**Step 1: Using Recursive Feature Machines to identify the most predictive features**

A Recursive Feature Machine (RFM) is a supervised machine learning model that incorporates feature learning into general non-parametric models through the Average Gradient Outer Product (AGOP) [37]. Unlike prior methods that used brute force (randomly selecting five variants $V_1$-$V_5$ to predict a target virus $V_0$, assessing that selection using cross-validation), RFM gives the feature importance of all variants so that the top candidates can be used to predict $V_0$. This leads to more efficiently identifying the predictive features, is not restricted to a pre-imposed number of features (*e.g.*, RFM does not always choose five features), and yields better predictions than a random search through a subset of possibilities (S12A Fig). For example, the Fig 3 analysis requires approximately 300 minutes using the brute force approach versus 20 minutes using CAPYBARA's RFM.

Given any differentiable predictor, $f$:$\mathbb{R}^d \rightarrow \mathbb{R}$ trained on $n$ data points $x^{(1)}, \ldots, x^{(n)} \in \mathbb{R}^d$, the AGOP operator $G(f)$ is the covariance matrix of the input-output gradients of the predictor over the training data,

$$G(f) = \frac{1}{n} \sum_{j=1}^{n} \nabla_x f(x^{(j)}) \, \nabla_x f(x^{(j)})^T \in \mathbb{R}^{d \times d}$$

(1)

This covariance captures the most predictive directions in its top eigenvectors, and the most important coordinates on its diagonal. RFM proceeded by obtaining an initial estimate of the target function using a standard kernel machine without feature learning. Given this initial estimate of the predictor, the AGOP of the predictor was computed on the training data, after which the inner product function was updated using the AGOP. RFM then recursed this procedure beginning with the transformed data. Formally, the algorithm proceeded as follows.

**Algorithm 1: Recursive Feature Machine (RFM)**

```
Inputs:
    •Training data:
        ◦x⁽¹⁾,..., x⁽ⁿ⁾∈Rᵈ: HAI of feature variants V₁-V_d for all n subjects
        ◦y ∈ Rⁿ: HAI of withheld virus-of-interest V₀ for each subject
    •Mahalanobis Laplace kernel k(x, x') = exp(-dist_M(x, x')/σ): Kernel function used to define
similarity between samples for kernel ridge regression. dist_M(x, x')=√((x-x')ᵀM (x-x')) is the
Mahalanobis distance with σ chosen by the median heuristic based on the training data, and positive
semidefinite matrix M∈ Rᵈ
    •T=5: Number of RFM iterations
    •μ=10⁻⁵: Ridge regularization coefficient, chosen to be small enough to avoid over-regularization
yet still stabilize kernel inversion
Steps:
    •Initialize feature matrix X₀ = [x⁽¹⁾,..., x⁽ⁿ⁾]ᵀ∈Rⁿˣᵈ
    •Initialize positive semidefinite matrix M=I_d
    •For t=0 to T-1:
        ◦Set α = (k_M(X_t, X_t) + μ I)⁻¹y
        ◦Define predictor, f⁽ᵗ⁾(x) = k_M(x, X_t) α
        ◦Update feature matrix M to be the AGOP, M = 1/n ∑_{j=1}ⁿ ∇_x f⁽ᵗ⁾(x⁽ʲ⁾) ∇_x f⁽ᵗ⁾(x⁽ʲ⁾)ᵀ ∈ Rᵈˣᵈ
Output: The diagonal elements of the learned feature transformation matrix (M_{T-1}) indicate the impor-
tance of each variant. Variants with diagonal values >0.1 are chosen as the predictive features for
target virus V₀.
```

**Step 2: Model training and internal error**

Within each external dataset, ridge regression models were trained on selected features identified by RFM. Note that RFM importance was not considered during ridge regression, since in the case of multiple degenerate but highly important features, only a single feature should be selected. Hyperparameters (ridge regularization strength, kernel bandwidth,

diagonal thresholds) were optimized via internal cross-validation (80% training, 20% validation splits), but were found to minimally vary (S12C Fig).

Following ridge regression, each variant feature with ridge coefficient >0.2 (in absolute value) was retained. When deriving universal cross-reactivity relations, if two studies predicted a target virus $V_0$ using the same variants as features, the ridge coefficients were averaged for each of the viruses in their equation.

## Step 3: Cross-study error calibration

Following prior work [38], to calibrate how accurately the model trained on study $S_j$ applied to study $S_0$, every possible virus $V_k \neq V_0$ was withheld one-by-one (in addition to excluding $V_0$) from both the training and testing datasets. 80% of sera in the training set were used to fit a ridge regression model in the training dataset, with the remaining sera used to compute the internal error $\sigma_{Internal}(V_k)$. All sera in the testing dataset were used to compute $\sigma_{External}(V_k)$. Performing this for all $V_k$ resulted in multiple points ($\sigma_{Internal}$, $\sigma_{External}$) that mapped the transferability of error between the two studies.

These paired internal–external errors were fit using a total-least-squares (orthogonal-distance) line, $\sigma_{External} = \alpha\,\sigma_{Internal} + \beta$. To account for the uncertainty of this fit (*i.e.*, highly scattered points with a poor best-fit line are more uncertain), we added to $\sigma_{External}$ the root-mean-square vertical distance of each point from the fitted line, $\delta = [1/m\sum_{k=1}^{m}(\alpha\,\sigma_{Internal}(V_0) + \beta - \sigma_{External}(V_k))^2]^{1/2}$. Lastly, the external error was forced to always be at least as large as the internal error. Altogether, the estimated error when predicting variant $V_0$ in $S_0$ is given by

$$\sigma_{Predict}(V_0) = \max(\alpha\,\sigma_{Internal}(V_0) + \beta + \delta,\ \sigma_{Internal}(V_0)) \tag{2}$$

## Step 4: Combining predictions from multiple datasets

When multiple studies $S_1$, $S_2$… predicted the HAI titers of virus $V_0$ in $S_0$, each subject had predictions $\mu_1 \pm \sigma_1$, $\mu_2 \pm \sigma_2$… (where $\sigma$ is a shorthand for $\sigma_{Predict}$). Predictions were combined using Bayesian weighting that is inversely proportional to predicted error squared, namely,

$$\frac{\sum_j \mu_j/\sigma_j^2}{\sum_j 1/\sigma_j^2} \pm \left(\frac{1}{\sum_j 1/\sigma_j^2}\right)^{1/2} \tag{3}$$

More confident predictions (smaller $\sigma_j$) are weighted more heavily, while highly inaccurate predictions ($\sigma_j \to \infty$) have little-to-no influence. As a result, all datasets can be included, and the algorithm will unbiasedly determine the most accurate predictions and use their values more heavily.

## Identification of HA1 residues underlying RFM similarity

To identify which HA1 sites best explain the RFM-derived virus-virus similarity, we performed iterative backward-elimination on the aligned HA1 head sequences (Method #1, *Minimal Key Residues*). At each iteration, every remaining position was temporarily removed, the edit-distance matrix was recomputed, and the position whose removal most decreased the Frobenius distance was eliminated. Iterations proceeded until removal no longer reduced the Frobenius norm (resulting in 28 sites). Robustness was assessed using one additional forward add-back (yielding 32 sites) and then backward pruning (31 sites). All HA1 positions are reported in H3 numbering, including the 16 amino-acid signal peptide.

## Software and computational resources

Analyses were implemented in Python using standard scientific libraries (NumPy, SciPy, scikit-learn). Code is available through the accompanying GitHub repository (https://github.com/TalEinav/CAPYBARA).

## Supporting information

**S1 Fig. Subject age distributions across datasets.** Datasets are ordered chronologically and by study group.
(TIF)

**S2 Fig. Prediction accuracy using all studies is consistently comparable to experimental noise.** Every other study in Table 2 is used to predict HAI titers for all variants in the study-of-interest (shown by the plot label).
(TIF)

**S3 Fig. Signed prediction error as a function of the measured ground truth titer.** Boxplots show median and inter-quartile range of prediction residual across all datasets, with sample counts annotated above each bin.
(TIF)

**S4 Fig. Prediction accuracy between every pair of studies.** (A) The estimated prediction accuracy for each pair of studies, computed as the mean $(1/\sigma_{Predicted})^2$ over all overlapping variants. Larger values indicate that the training dataset will be weighted more heavily in combined-study predictions. (B) Signed prediction error on log titers, $\log_2$(measured HAI/predicted HAI) for all variants in each pair of studies. Red indicates that measured titers were larger than predicted titers on average. Training is either done using all studies (top row) or using a single study (all other rows).
(TIF)

**S5 Fig. Distribution of errors for individual and combined predictions.** Fold-error ($\sigma_{Actual}$) of predictions for every subject and virus using (A) each dataset to make a separate prediction and (B) all datasets to make combined predictions. Red shading marks the region of ≤4x error, and the annotated percentages indicate the fraction of predictions that fall within this threshold.
(TIF)

**S6 Fig. Combined predictions outperform averaged predictions from individual studies.** Prediction errors ($\sigma_{Actual}$) for all viruses in all infection studies were computed using two other datasets for training. These two datasets either independently predicted each virus, and their resulting predictions were averaged [x-axis] or CAPYBARA was used to combine these predictions by more heavily weighing the dataset that was more similar to the target infection study [y-axis]. Points below the diagonal indicate improved performance with the combined model.
(TIF)

**S7 Fig. Predicting HAI responses across other influenza subtypes.** (A) Heatmap of the predicted vs measured RMSE ($\sigma_{Actual}$) across all subjects and overlapping variants for H1N1 (left column), B Victoria (middle column), and B Yamagata (right column). Within each heatmap, training is either done using all studies (top row) or using a single study (all other rows). (B-C) All predicted vs measured HAIs when training on (B) a single study or (C) all other studies. The number $N$ of predictions is larger for pairwise predictions since the same serum-virus pair is predicted multiple times using different training datasets. The diagonal line $y = x$ represents perfect predictions.
(TIF)

**S8 Fig. Predictions remain robust with the inclusion of a noisy dataset.** (A) Example predictions from Fig 2A using an individual dataset (left and middle columns) or the combination of both datasets (right column) to predict titers in 2017 UGA$_{Vac}$. Labels above each plot identify the training→testing dataset. (B) A study with random data predicted this same testing dataset either individually (left) or in addition with the other two training datasets from Panel A (right).
(TIF)

**S9 Fig. Multiplying all HAI titers in one study does not significantly change prediction accuracy.** Predicted versus measured HAI titers for two representative vaccine studies based on all other studies. Titers are predicted in (A) 2010

Fonv$_\text{Vac}$ and (B) 2016 Fox$_\text{HCW,Vac}$ using the original data (*left*) or after multiplying all titers in that single study by 4x (*right*) to demonstrate the effects of one study having systematically higher titers.
(TIF)

**S10 Fig. Feature importance via RFM.** The importance of each virus feature (column) when predicting a target virus ($V_0$, row). Feature importance is quantified within a single study. Only viruses with feature importance≥0.1 shown, as these viruses are subsequently used in ridge regression when predicting the target virus. Any virus not picked is shown in white.
(TIF)

**S11 Fig. Identification of HA1 head positions that best reproduce RFM-derived virus-virus similarity.** (A-B) Arc plots connecting each virus to its predictive partners. (A) Arcs colored by raw sequence distance computed from the selected 31 positions. (B) Arcs colored by the sequence distance computed from the 167 positions selected from virus pairs 10+ years apart and with high importance (≥0.7) in 80% of pairs. (C) Example sequence H3N2 A/Perth/16/2009 with canonical epitopes denoted by different colored lines above each position, and key residues found from two methods in this study highlighting the corresponding position numbers. (D) List of the 31 HA1 amino acids leading to the minimum Frobenius norm, annotated by which H3N2 epitope they fall into. Amino acid positions include the 16 amino-acid signal peptide.
(TIF)

**S12 Fig. CAPYBARA achieves lower prediction error than brute force approaches and allows for predictive uncertainty estimation.** (A) Comparison of fold-error for pairwise models generated by brute-force selection (running ridge regression on five randomly selected viruses, repeating 50 times to find the best five viruses) versus CAPYBARA (runs RFM a single time to identify the most predictive features and then ridge regression). Each point represents an overlapping virus between each dataset pair. More points lie above the diagonal and the average error is slightly smaller along the *x*-axis, with both traits indicating better performance with CAPYBARA. (B) Predicted versus actual error across all datasets using CAPYBARA, with each point representing all measurements for one virus in one study. We expect the predicted error to represent an upper bound, worst case error ($\sigma_\text{Actual} \leq \sigma_\text{Predict}$), which is satisfied in the vast majority of cases. (C) Heatmap of mean $\sigma_\text{Actual}$ across all dataset pairs for different hyperparameter settings for the diagonal threshold and bandwidth in RFM, showing nearly comparable prediction accuracy across all parameter choices.
(TIF)

**S1 File.**
(XLSX)

## Acknowledgments

We especially thank the experimental groups who shared their data, and we hope this paper will inspire other groups to integrate their datasets for everyone's benefit. We always welcome pointers to new datasets. We further acknowledge Adit Radha and Mikhail Belkin for useful discussions.

## Author contributions

**Conceptualization:** Sierra Orsinelli-Rivers, Tal Einav.

**Methodology:** Sierra Orsinelli-Rivers, Daniel Beaglehole.

**Writing – original draft:** Sierra Orsinelli-Rivers, Tal Einav.

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
