## [Decision Letter · Decision Letter 0]

23 Sep 2025

CAPYBARA: A Generalizable Framework for Predicting Serological Measurements Across Human Cohorts

PLOS Computational Biology

Dear Dr. Einav,

Thank you for submitting your manuscript to PLOS Computational Biology. After careful consideration, we feel that it has merit but does not fully meet PLOS Computational Biology's publication criteria as it currently stands. Therefore, we invite you to submit a revised version of the manuscript that addresses the points raised during the review process.

Please submit your revised manuscript within 60 days Nov 23 2025 11:59PM. If you will need more time than this to complete your revisions, please reply to this message or contact the journal office at ploscompbiol@plos.org. Please include the following items when submitting your revised manuscript:

We look forward to receiving your revised manuscript.

Kind regards,

Tyler Cassidy

Academic Editor

PLOS Computational Biology

James Faeder

Section Editor

PLOS Computational Biology

**Additional Editor Comments (if provided):**

In general, the reviewers appreciated this manuscript. However, they raised important concerns that must be addressed before publication (specifically, both major points from Reviewer 3 and the final two points from Reviewer 2). Further, as mentioned by reviewer 1 and 4, the manuscript may benefit from a discussion of how this approach may apply to vaccine design.

**Journal Requirements:**

At this stage, the following Authors/Authors require contributions: Daniel Beaglehole, Sierra Orsinelli-Rivers, and Tal Einav. Please ensure that the full contributions of each author are acknowledged in the "Add/Edit/Remove Authors" section of our submission form.

5) We notice that your supplementary Figures are included in the manuscript file. Please remove them and upload them with the file type 'Supporting Information'. Please ensure that each Supporting Information file has a legend listed in the manuscript after the references list.

7) Please send a completed 'Competing Interests' statement, including any COIs declared by your co-authors. If you have no competing interests to declare, please state "The authors have declared that no competing interests exist". Otherwise please declare all competing interests beginning with the statement "I have read the journal's policy and the authors of this manuscript have the following competing interests"

**Reviewers' comments:**

Reviewer's Responses to Questions

**Comments to the Authors:**

Reviewer #1: CAPYBARA: A Generalizable Framework for Predicting Serological Measurements Across Human Cohorts

This manuscript develops an algorithm to predict HAI titers to a large number of variants using serological measures of only a few variants, by exploiting cross-reactivity information from historical data sets. A significant limitation is that the variant of interest must already exist in the historical data set, and the need to test at least some variants in the current study — unlike ab initio models based on viral sequence like in Loes et al (Ref #6). The CAPYBARA algorithm could potentially be useful in reducing cost and labor, and for understanding the landscape of cross-reactivity, but the latter is not fully explored. Specific commons and questions:

• The algorithm is rather ad-hoc in nature, combing approaches from ML (RFM), frequentist statistics (ridge regression), and Bayesian statistics (study weighting). Depending on the reader’s preference for methodological consistency, this is either a trivial or serious issue.

• I could not get the GitHub notebook to work directly — for example, the torchmetrics package is not specified in requirments.txt so the import statement failed. Even after fixing that, other import errors arose. This was done in a virtual environment with python 3.10 as suggested, so environmental conflicts are not likely to be the issue.

• It does not appear that batch correction or “normalization” methods were applied to the data sets from different studies. Can you explain why this is not feasible or necessary or useful?

• The term “feature” is ubiquitous but not very well-defined. It appears to refer to the HAI titer of a variant, but this is easily missed by the reader. Can you provide a short description of the features used to help the reader?

• Since other features like “age” and “year of study” influence prediction accuracy, why not include these as additional features in CAPYBARA?

• Can new vaccine/infection studies with HAI data be easily added to CAPYBARA using Bayesian updating, or must the algorithm be re-trained?

• Can you provide a clearer explanation of what CAPYBARA revealed about the cross-reactivity landscape? For example, does it provide useful guidance for designing the next seasonal or universal influenza vaccine?

Reviewer #2: Please see attached comments

Reviewer #3: The manuscript by Orsinelli-Rivers describes the development of an analytical workflow for modeling and mapping the relationships among different Influenza virus serological datasets. I commend the authors for tackling this important problem and for providing an initial framework to infer influenza virus cross-reactivity from existing data. However, an important issue not addressed in the study is the accuracy of the model’s predictive power in relation to antigenic changes. Notably, one of the main conclusions is that predictions are accurate when datasets are derived from studies conducted less than 5 years apart. This inherently suggests that serological assays performed on antigenically similar viruses, typically those within a 5-year window, are more likely to yield more accurate cross-reactivity predictions. Another important point not addressed in the study is the generalizability of this analytical workflow, despite this being emphasized in both the abstract and the title of the manuscript. These and other issues included below, should be addressed before the manuscript can be considered for further evaluation.

Major comments:

1. While the analytical workflow is useful for predicting serological cross-reactivity among H3N2 influenza viruses, it has a major limitation regarding the biological interpretation of antigenic differences between the viruses assessed. As shown in Figure 4, the accuracy of predictions was influenced by the time elapsed between the datasets used for analysis and the variants being evaluated—viruses within a five-year window were more likely to yield accurate cross-reactivity predictions. This likely reflects lower antigenic divergence within such a timeframe; in other words, H3 hemagglutinins (HAs) within five years of each other are generally more antigenically similar, with this similarity declining beyond that period. This trend is clearly visible in Figure 5 and Supplementary Figure S5. However, there are notable exceptions—some viruses that are many years apart still produced accurate predictions. This discrepancy is not addressed in the study. It would be helpful for the authors to explore this further, possibly by analyzing HA sequences to identify amino acid residues that may underlie cross-reactivity between antigenically distant viruses. An amino acid sequence alignment could provide insights into this issue. While it is understandable that the primary focus of the manuscript is the development of the analytical workflow, the biological interpretation of the results should not be overlooked. Including such analysis in future iterations of the tool could improve its utility and generalizability. This is particularly important given the history of antigenic shifts in influenza viruses—for example, the 2009 H1N1 pandemic virus exhibited close antigenic similarity to strains that circulated prior to the 1968 H3N2 pandemic, highlighting the complexity of antigenic evolution across time.

2. While the authors mention that this analytical tool is generalizable, they did not provide evidence that the workflow works effectively for other influenza subtypes (e.g. H1N1) or other viruses. Therefore, it would be more appropriate to avoid making this assumption in the manuscript. I would also recommend a slight revision of the tittle to better reflect the specific scope and finding of the study: “CAPYBARA: A Framework for Predicting H3N1 influenza Serological Measurements Across Human Cohorts”.

Minor Comments:

1. In the introduction, the authors should indicate that the study was performed only with H3N2 viruses.

2. The second paragraph of the introduction indicates “While thousands of new variants (or strains) emerge each year, …”. Please provide a reference for this statement.

3. The fifth paragraph from the introduction, which starts with: “A key innovation…” provides discussion points of the workflow, and therefore this should be moved to the discussion section.

4. I am a bit confused by the last sentence of the introduction, as the authors did not specifically quantified differences among study populations, experimental conditions, and virus panels. Did they mean to say that, despite differences among study populations, experimental conditions, and virus panels, the analyses workflow allows to estimate cross-reactivity among different Influenza virus serological datasets? I suggest revising this sentence. Also edit the beginning of the sentence to read: “By predicting these interactions for H3N2 viruses, …”

5. In the second paragraph, line 2 of the discussion, did the authors meant to say “…cross-reactivity across influenza variants.”?

6. In the fourth paragraph of the discussion, did the authors meant to say “…didn’t require”?

Reviewer #4: I find this work interesting and I think the manuscript is for the most part well-written.

There is a sentence, page 8 of the manuscript, "Prediction accuracy can only decrease..." Please clarify this statement, it is vague saying adding more datasets when the number of datasets is not restricted. It lists studies {S1, S2,...} Perhaps this is intentionally left vague because we cannot easily say an exact number of datasets to stop at, however the statement I believe can be made more clear. Later, in the paper there is again mention of very noisy datasets having little impact and it also states all datasets can be included, which seems to go against the statement where adding more datasets will decrease accuracy. Hence, some clarification of these statements would be beneficial to the reader.

On page 16 of the manuscript, "The algorithm did require..." It seems you meant to write didn't or did not.

I would be interested in seeing this methodology utilized in other studies for vaccine development. I think with some minor writing revisions this is an acceptable paper.

**Have the authors made all data and (if applicable) computational code underlying the findings in their manuscript fully available?**

The PLOS Data policy requires authors to make all data and code underlying the findings described in their manuscript fully available without restriction, with rare exception (please refer to the Data Availability Statement in the manuscript PDF file). The data and code should be provided as part of the manuscript or its supporting information, or deposited to a public repository. For example, in addition to summary statistics, the data points behind means, medians and variance measures should be available. If there are restrictions on publicly sharing data or code —e.g. participant privacy or use of data from a third party—those must be specified.requires authors to make all data and code underlying the findings described in their manuscript fully available without restriction, with rare exception (please refer to the Data Availability Statement in the manuscript PDF file). The data and code should be provided as part of the manuscript or its supporting information, or deposited to a public repository. For example, in addition to summary statistics, the data points behind means, medians and variance measures should be available. If there are restrictions on publicly sharing data or code —e.g. participant privacy or use of data from a third party—those must be specified.requires authors to make all data and code underlying the findings described in their manuscript fully available without restriction, with rare exception (please refer to the Data Availability Statement in the manuscript PDF file). The data and code should be provided as part of the manuscript or its supporting information, or deposited to a public repository. For example, in addition to summary statistics, the data points behind means, medians and variance measures should be available. If there are restrictions on publicly sharing data or code —e.g. participant privacy or use of data from a third party—those must be specified.requires authors to make all data and code underlying the findings described in their manuscript fully available without restriction, with rare exception (please refer to the Data Availability Statement in the manuscript PDF file). The data and code should be provided as part of the manuscript or its supporting information, or deposited to a public repository. For example, in addition to summary statistics, the data points behind means, medians and variance measures should be available. If there are restrictions on publicly sharing data or code —e.g. participant privacy or use of data from a third party—those must be specified.

Reviewer #1: Yes

Reviewer #2: Yes

Reviewer #3: None

Reviewer #4: Yes

PLOS authors have the option to publish the peer review history of their article (what does this mean?). If published, this will include your full peer review and any attached files.). If published, this will include your full peer review and any attached files.). If published, this will include your full peer review and any attached files.). If published, this will include your full peer review and any attached files.

...

Reviewer #1: No

Reviewer #2: No

Reviewer #3: No

Reviewer #4: No

**Figure resubmission:**
---

## [Decision Letter · Decision Letter 1]

16 Mar 2026

Dear Dr. Einav,

We are pleased to inform you that your manuscript 'CAPYBARA: A Generalizable Framework for Predicting Serological Measurements Across Human Cohorts' has been provisionally accepted for publication in PLOS Computational Biology.

Best regards,

Tyler Cassidy

Academic Editor

PLOS Computational Biology

James Faeder

Section Editor

PLOS Computational Biology

Reviewer's Responses to Questions

**Comments to the Authors:**

Reviewer #1: I have reviewed the revised manuscript. The authors have done a thorough job addressing reviewer comments and the GitHub code is now executable. I am happy to recommend Acceptance of the manuscript.

Reviewer #2: Please see my attachment.

**Have the authors made all data and (if applicable) computational code underlying the findings in their manuscript fully available?**

The PLOS Data policy requires authors to make all data and code underlying the findings described in their manuscript fully available without restriction, with rare exception (please refer to the Data Availability Statement in the manuscript PDF file). The data and code should be provided as part of the manuscript or its supporting information, or deposited to a public repository. For example, in addition to summary statistics, the data points behind means, medians and variance measures should be available. If there are restrictions on publicly sharing data or code —e.g. participant privacy or use of data from a third party—those must be specified.requires authors to make all data and code underlying the findings described in their manuscript fully available without restriction, with rare exception (please refer to the Data Availability Statement in the manuscript PDF file). The data and code should be provided as part of the manuscript or its supporting information, or deposited to a public repository. For example, in addition to summary statistics, the data points behind means, medians and variance measures should be available. If there are restrictions on publicly sharing data or code —e.g. participant privacy or use of data from a third party—those must be specified.requires authors to make all data and code underlying the findings described in their manuscript fully available without restriction, with rare exception (please refer to the Data Availability Statement in the manuscript PDF file). The data and code should be provided as part of the manuscript or its supporting information, or deposited to a public repository. For example, in addition to summary statistics, the data points behind means, medians and variance measures should be available. If there are restrictions on publicly sharing data or code —e.g. participant privacy or use of data from a third party—those must be specified.requires authors to make all data and code underlying the findings described in their manuscript fully available without restriction, with rare exception (please refer to the Data Availability Statement in the manuscript PDF file). The data and code should be provided as part of the manuscript or its supporting information, or deposited to a public repository. For example, in addition to summary statistics, the data points behind means, medians and variance measures should be available. If there are restrictions on publicly sharing data or code —e.g. participant privacy or use of data from a third party—those must be specified.

Reviewer #1: Yes

Reviewer #2: Yes

PLOS authors have the option to publish the peer review history of their article (what does this mean?). If published, this will include your full peer review and any attached files.). If published, this will include your full peer review and any attached files.). If published, this will include your full peer review and any attached files.). If published, this will include your full peer review and any attached files.

...

Reviewer #1: No

Reviewer #2: No

---

## [Editor Report · Acceptance letter]

PCOMPBIOL-D-25-01204R1

CAPYBARA: A Generalizable Framework for Predicting Serological Measurements Across Human Cohorts

Dear Dr Einav,

I am pleased to inform you that your manuscript has been formally accepted for publication in PLOS Computational Biology. Your manuscript is now with our production department and you will be notified of the publication date in due course.

With kind regards,

Anita Estes
